

# Skeletal variation in bird domestication: limb proportions and sternum in chicken, with comparisons to mallard ducks and Muscovy ducks

Carlos Manuel Herrera-Castillo[1], Madeleine Geiger[1], Daniel Núñez-León[1], Hiroshi Nagashima[2], Sabine Gebhardt-Henrich[3], Michael Toscano[3] and Marcelo R. Sanchez-Villagra[1]

[1] Palaeontological Institute and Museum, University of Zurich, Zurich, Switzerland
[2] Division of Gross Anatomy and Morphogenesis, Niigata University Graduate School of Medical and Dental Sciences, Niigata, Japan
[3] Center for Proper Housing: Poultry and Rabbits (ZTHZ), Animal Welfare Division, Veterinary Public Health Institute, University of Bern, Bern, Switzerland

Corresponding author
Marcelo R. Sanchez-Villagra,
m.sanchez@pim.uzh.ch

## ABSTRACT

**Background**. Domestication, including selective breeding, can lead to morphological changes of biomechanical relevance. In birds, limb proportions and sternum characteristics are of great importance and have been studied in the past for their relation with flight, terrestrial locomotion and animal welfare. In this work we studied the effects of domestication and breed formation in limb proportions and sternum characteristics in chicken (*Gallus gallus*), mallard ducks (*Anas plathyrhynchos*) and Muscovy ducks (*Cairina moschata*).

**Methods**. First, we quantified the proportional length of three long bones of the forelimb (humerus, radius, and carpometacarpus) and the hind limb (femur, tibiotarsus, and tarsometatarsus) in domestic chickens, mallard ducks, and Muscovy ducks and their wild counterparts. For this, we took linear measurements of these bones and compared their proportions in the wild *vs.* the domestic group in each species. In chicken, these comparisons could also be conducted among different breeds. We then evaluated the proportional differences in the context of static and ontogenetic allometry. Further, we compared discrete sternum characteristics in red jungle fowl and chicken breeds. In total, we examined limb bones of 287 specimens and keel bones of 63 specimens.

**Results**. We found a lack of significant change in the proportions of limb bones of chicken and Muscovy duck due to domestication, but significant differences in the case of mallard ducks. Variation of evolvability, allometric scaling, and heterochrony may serve to describe some of the patterns of change we report. Flight capacity loss in mallard ducks resulting from domestication may have a relation with the difference in limb proportions. The lack of variation in proportions that could distinguish domestic from wild forms of chicken and Muscovy ducks may reflect no selection for flight capacity during the domestication process in these groups. In chicken, some of the differences identified in the traits discussed are breed-dependent. The study of the sternum revealed that the condition of crooked keel was not unique to domestic chicken, that some sternal characteristics were more frequent in certain chicken breeds than in others,
and that overall there were no keel characteristics that are unique for certain chicken breeds. Despite some similar morphological changes identified across species, this study highlights the lack of universal patterns in domestication and breed formation.

# INTRODUCTION

Through domestication, much morphological diversity has been generated (*Darwin, 1868*; *Herre & Röhrs, 1990*). Domesticated animals exhibit phenotypic changes compared to their wild counterparts (*Clutton-Brock, 1999*). This is the case in domestic birds, as shown in recent works on the skulls of pigeons (*Columba livia*) and chickens (*Gallus gallus*) (*Young et al., 2017*; *Stange et al., 2018*) and the integument in chickens (*Núñez León et al., 2019*). In contrast, postcranial anatomy, including limb bones, has rarely been dealt with analytically and globally in domesticated species (see *Wayne, 1986* for dogs; *Van Grouw, 2018* for an overview for many species), despite having been widely studied in bird evolutionary biology (*Middleton & Gatesy, 2000*; *Dyke & Nudds, 2009*; *Nudds et al., 2013*). *Darwin (1868)* mentioned that changes in limb length (or lack thereof) in domestic birds can be observed. Especially notable is the reduction of wing length of the mallard duck (*Anas plathyrhynchos*) due to domestication (*Darwin, 1868*).

Limb proportions can be of importance in biomechanical processes such as flight ability. *Middleton & Gatesy (2000)* highlighted the importance of the humerus relative proportion in the forelimb, where flightless birds have relatively longer humeri while flight manoeuvrability is being linked to shorter humeri. The selective pressure for flight is thus expected to influence the relative proportion of the different bones in the forelimb (*Mason, 1984*). While the wild type red junglefowl (RJF) is generally considered a short distance flier and a non-migratory bird (*Bird Life International, 2016*), flight capability in its domestic counterpart, the chicken, is breed-dependent (*Schippers, Simons & Kippen, 2013*). Certain chicken breeds are known to have only limited flight capacity or even completely lost their flight capacity (*e.g.*, ukokkei and Polish breeds) and this is largely attributed to the unfitness of their plumage for such task (*Ekarius, 2007*; *León et al., 2021*). Differences in flight capacity have also been reported depending on the weight of the breed: the heavier, the less prone to sustained flight (*Darwin, 1868*; *Schippers, Simons & Kippen, 2013*).

It is generally agreed that domestic mallard ducks have lost their flight capacity when compared to their wild counterpart, which in contrast to the RJF, is a migratory bird (*Accordi & Barcellos, 2006*; *Bird Life International, 2016*). This change due to domestication has been attributed to a gain in weight and reduction in the proportional wing length (*Darwin, 1868*; *Cnotka, 2006*). Muscovy ducks (*Cairina moschata*) are non-migratory birds (*Bird Life International, 2018*), but even the domestic forms retain their flight ability (*Swatland, 1980*). Unlike the case of differences in wing length between domestic and wild

mallard ducks, finding osteological differences between domestic and wild Muscovy ducks has proven more difficult (*Angulo, 1998*; *Stahl, Muse & Delgado-Espinoza, 2006*).

Another skeletal element of biomechanical importance in birds is the sternum, the site of attachment of the pectoralis muscle, important for producing mechanical work during downstroke and pronating the wing, and the supracoracoideus muscle, which elevates and supinates the wing during upstroke (*Biewener, 2011*). The sternum also fulfils an important role in respiration as it is the site of attachment of m. obliquus abdominis externus and m. rectus abdominis (*Lowi-Merri et al., 2021*).

The sternum varies in shape and proportions and is a source of intensive study in poultry research concerned with welfare due to widespread reports of damage ranging between 30 and 100% of commercial hens (*Buckner et al., 1949*; *Fleming et al., 2004*; *Harlander-Matauschek et al. , 2015*; *Toscano et al., 2020*). Keel damage may include crookedness or bending, as well as fractures (*Casey-Trott et al., 2015*) and has been observed to occur in different proportions between breeds—potentially also related to wing loading—and between sexes (*Kittelsen et al., 2021*; *Stratmann et al., 2016*). Deformations of the keel can be divided into two types: fractures, defined as sharp bends, fragmented sections or shearing of the keel with or without thickened bone (callus) (*Fleming et al., 2004*; *Eusemann et al., 2018*; *Tracy et al., 2019*; *Thøfner et al., 2020*); and deviations, defined as abnormally shaped bone containing section(s) that vary from an ideally perfect two-dimensional straight plane that has not resulted from fracture (*Casey-Trott et al., 2015*). Pathological characterization of keel bone fractures in laying hens does not support external trauma as the underlying cause (*Thøfner et al., 2020*). Therefore, here we use the term 'crookedness' to denote exclusively the deviations, not fractures, of the keel, as defined in *Casey-Trott et al. (2015)*. The problem appears somewhat historical as it was referred to by *Darwin* (*1868*, p. 282) as 'generally so much deformed that it is scarcely possible to compare its form strictly in the several breeds'. The crookedness of the keel has been largely studied in the context of its causes, its relation with animal welfare, and its effect on productivity (*Blount, 1933*; *Warren, 1937*; *Waters, 1949*; *Buckner et al., 1949*; *Fleming et al., 2004*; *Eusemann et al., 2018*; *Thøfner et al., 2020*) and the market value of the domestic chicken (*Hyre, 1995*). Keel crookedness has been reported to have environmental and genetic origins, appearing mostly when chickens are given the opportunity to perch (*Blount, 1933*; *Warren, 1937*; *Waters, 1949*; *Buckner et al., 1949*; *EFSA, 2015*; *Gebhardt-Henrich, Toscano & Würbel, 2017*). However, whether crookedness of the keel and other alterations of sternum characteristics typically found in domestic chicken are associated with domestication in its initial, less intense phase (*Vigne, 2011*), or if it appeared only with breed formation and improvement has not been determined (*Blount, 1933*; *Warren, 1937*; *Waters, 1949*; *Hyre, 1995*; *Fleming et al., 2004*; *Kittelsen et al., 2021*). We would expect that wild forms exhibit less deformed and more compact keel bones, *i.e.,* straight and non-notched carina sterni and smooth foramen pneumaticum and pars cardiaca, related with more pronounced use and therefore ossification of the bone. Deformation and increased sponginess, on the other hand, might be related to body size changes due to domestication and breed formation.

In this study, we analysed the limb proportions of three domestic bird species: chicken (*G. gallus*), mallard duck (*A. plathyrhynchos*), and Muscovy duck (*C. moschata*) in order

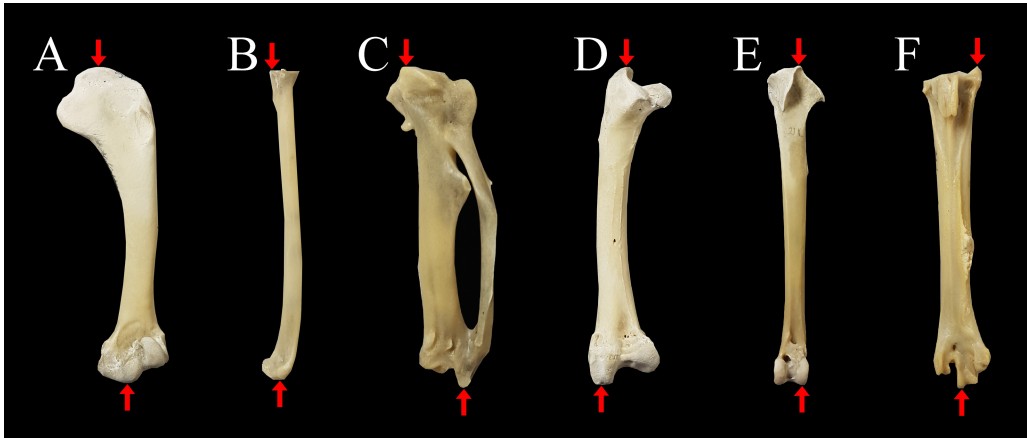

**Figure 1 Measurements of limb bones taken in the current study.** Anatomical structures used to define measurements parallel to the longitudinal axis of the limb bones (indicated with red arrows) include: (A) humerus length from the most proximal point of the caput humeri (upper arrow) to the most distal point of the condylus ventralis (lower arrow); (B) radius length from the most proximal point of the cotyla humeralis (upper arrow) to the most distal point of the facies articularis radiocarpalis (lower arrow); (C) carpometacarpus length from the most proximal point of the trochlea carpalis (upper arrow) to the most distal point of the facies articularis digitalis minor (lower arrow); (D) femur length from the most proximal point of the trochanter femoris (upper arrow) to the most distal part of the condylus lateralis (lower arrow); (E) tibiotarsus length from the most proximal part of the crista cnemialis cranialis (upper arrow) to the most distal point of the epicondylus lateralis (lower arrow); (F) tarsometatarsus length from the most proximal point of the crista medialis hypotarsi (upper arrow) to the most distal point of the trochlea metatarsi III (lower arrow). Specimen is an Araucana chicken housed at the Palaeontological Institute and Museum, University of Zurich: PIMUZ A/IV 163.

to determine whether there have been changes due to domestication and breed formation. We examined this matter in the context of evolvability, allometry, heterochrony, and selective pressures (artificial selection) that affect flight capacity. In the case of chickens, we examined these changes also among a diversity of breeds. Further, we investigated variation in the sternum by coding discrete anatomical features that characterize its variation between wild (RJF) and domestic chicken, as well as among domestic chicken breeds. Given the previous studies on differences between wild and domestic animals and the importance in locomotion of forelimb and hind limb proportions and sternum characteristics, we aimed to determine whether differences exist between wild and domestic forms of three bird species and to shed light on the possible underlying patterns.

## MATERIALS & METHODS

### Limb bone measurements and sternum character states

We took measurements of the length of the humerus, radius and carpometacarpus for the forelimb and length of the femur, tibiotarsus and tarsometatarsus for the hind limb. For each bone, we measured the length as the maximum length of the bone parallel to the measuring device (Fig. 1), using a digital caliper (Electronix Express 0604CAL6++ LCD Digital Calipers) to the nearest tenth of a mm. All measurements were conducted by one observer (CMHC).
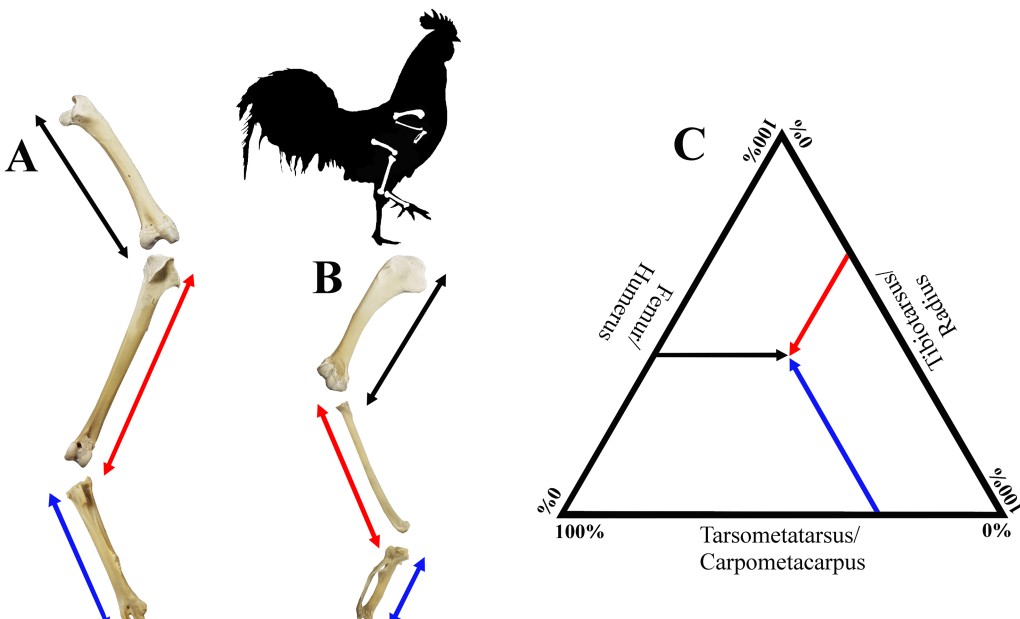

**Figure 2   Visualisation of estimates of limb length of the forelimb (A) and the hind limb (B), as well as limb bone proportions in a ternary plot (C).** Hind limb length (A), was defined as the sum of the lengths of the femur (black), tibiotarsus (red) and tarsometatarsus (blue); forelimb length (B), was defined as the sum of the lengths of the humerus (black), radius (red) and carpometacarpus (blue). See Fig. 1 for detailed descriptions of measurements. A ternary plot is used to show the proportions for each limb as a percentage of the whole limb length (C).

The total length of the forelimb was here defined as the sum of the lengths of the carpometacarpus, radius and humerus, while the total length of the hind limb was defined as the sum of the lengths of the tarsometatarsus, tibiotarsus and femur (Fig. 2). Although the thus estimated limb length does not equal 'functional limb length' because cartilage, soft tissue, and many autopodial elements were not considered, this estimate serves as an approximation (*e.g.*, *Zeffer, Johansson & Marmebro, 2003*). We considered the proportion of a bone to be the length of that bone with respect to the total length of the limb. This ratio was calculated by dividing the length of the bone by the total length of the respective limb.

For the analysis of feature variations in the sternum in chicken, we accounted for five characters (Fig. 3): (1) carina sterni is defined as 'straight' when the direction of growth is maintained in a straight line, as opposed to 'crooked', in which the growth is tortuous with one or multiple pronounced torsions; (2) carina sterni is defined as 'notched' when the direction of the lower margin of the carina in lateral view changes direction, creating a notch (it is defined as 'non notched' if this does not happen); (3) pars cardiaca and foramen pneumaticum are described as 'spongy' or 'smooth' depending on the type of external appearance, either porous or flat, respectively; (4) the caudal end of the trabecula mediana of the sternum is differentiated between 'fanned' (end wider than the rest of the

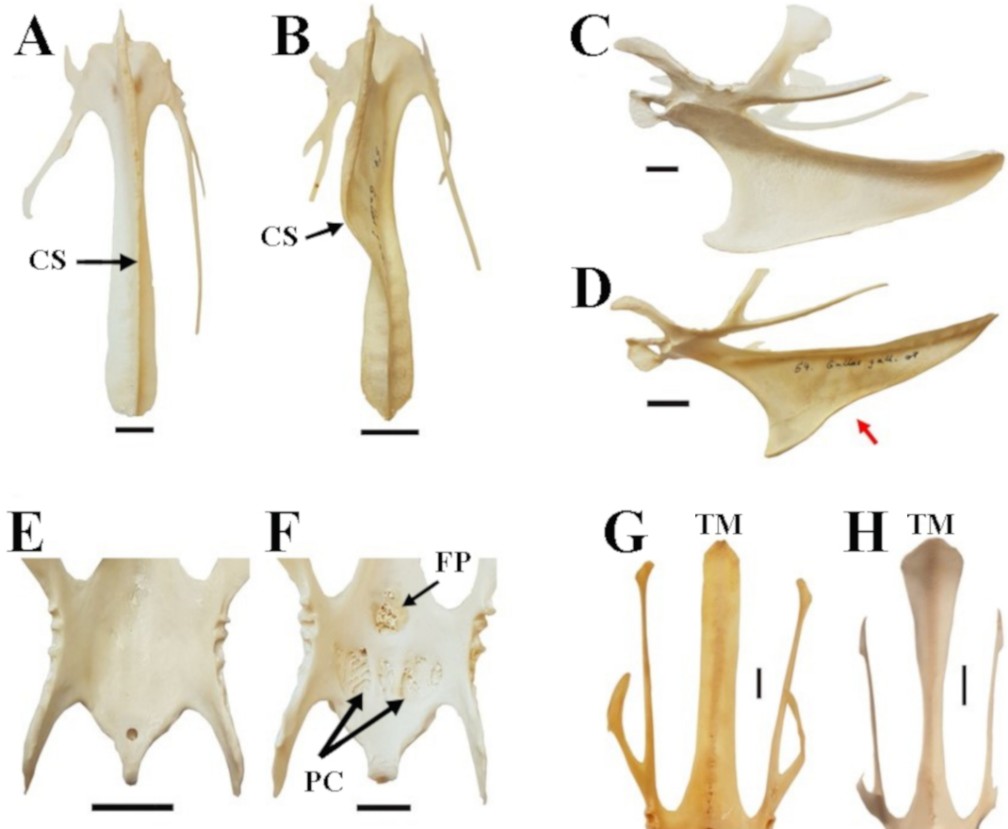

**Figure 3** **Characters and character states of the sternum in chicken as used in this study.** (A) Straight carina sterni (CS) (Araucana SAPM-AV-05259); (B) crooked carina sterni (RJF SAPM-AV-05218); (C) non-notched carina sterni (Araucana SAPM-AV-05274); (D) notched carina sterni (RJF SAPM-AV-05218), where the notch is indicated by the red arrow; (E) smooth surface and poorly defined concavities of pars cardiaca (PC) and foramen pneumaticum (FP) (RJF SAPM-AV-05234); (F) spongy, well-defined concavities of PC and FP (Araucana SAPM-AV-05259)); (G) Straight end of the trabecula mediana (TM) in dorsal view (Araucana SAPM-AV-05264); (H) fanned end of the TM in dorsal view (RJF SAPM-AV-05266). Specimens are form the Staatssammlung für Anthropologie und Paläoanatomie München (Germany). Scale = 1 cm.

trabecula in dorsal view) and 'straight' (end as wide as the rest of the trabecula in dorsal view).

## Specimens

We compared limb proportions in the wild *vs.* domestic forms in three bird species: mallard duck *(A. platyrhynchos)*, Muscovy duck *(C. moschata),* and chicken *(G. gallus)*. We chose these three species for their historical importance in bird domestication (*e.g.*, *Larson & Fuller, 2014*) and wide availability in osteological collections where the labelling as wild and domestic forms is reliable. Affiliation to the different species, and within the species to the wild or the domestic group and the breed (if available), was determined by means of the collection labels and databases. Specimens were only included if the collection labels clearly identified them as wild or domestic. The RJF was used here as the wild relative of the

**Table 1 Chicken breeds and number of specimens measured in this study.** Although the vast majority of specimens were complete, a few specimens lack forelimb or hind limb measurements (see also File S1).

| Breed | $N_{Forelimb}$ | $N_{Hindlimb}$ | Breed | $N_{Forelimb}$ | $N_{Hindlimb}$ |
|---|---|---|---|---|---|
| Red junglefowl | 30 | 29 | Light sussex | 3 | 2 |
| 'African cock' | 1 | 1 | Pennsylvania naked neck | 2 | 2 |
| Appenzeller barthuhn | 3 | 3 | Rhode Island | 1 | 0 |
| Appenzeller Spitzhaubenhuhn | 5 | 5 | Rumpless fowl | 1 | 1 |
| Araucana | 13 | 13 | Shamo | 8 | 8 |
| Bantam | 7 | 8 | Silver pencil hamburgh hen | 1 | 0 |
| Brahmahuhn | 1 | 1 | Polish | 5 | 5 |
| Burmese bantam | 1 | 1 | Spanish cock | 2 | 2 |
| Chabo | 16 | 16 | Sultan | 2 | 1 |
| Cochin | 6 (3 standard, 3 dwarf) | 6 (3 standard, 3 dwarf) | Tail-less bantam | 1 | 1 |
| Dorking cock | 1 | 1 | Ukokkei | 15 | 15 |
| Game | 2 | 2 | Welsumer | 3 | 2 |
| Gold spangle Polish | 1 | 1 | White crested rumpless sultan | 1 | 1 |
| Golden pencil hamburgh | 1 | 1 | White dorkinghen | 1 | 1 |
| Italiener | 1 | 1 | White leghorn | 4 | 4 |
| Kulm | 1 | 1 | Zwergwelsumer | 2 | 2 |

domestic chicken, although there has also been introgression of grey junglefowl (*Al-Nasser et al., 2007*; *Eriksson et al,, 2008*; *Lawal et al., 2020*). In all species, both sexes were integrated in the study. To reduce the influence of ontogenetic variation we considered only adult, *i.e.*, skeletally mature specimens for this work, with individuals considered adults if the epiphyses of all long bones were externally completely fused to the diaphyses.

For the mallard duck, 54 wild and 34 domestic specimens were measured. Of the 34 domestic mallards, nine were from known breeds (four aylesbury, two black labrador or buenos aires drake, one khaki campbell, and two white calldrake). Given the lack of breed information for most specimens, we did not consider the breeds separately. For the Muscovy duck, 39 wild and 17 domestic specimens were measured. Also in this species, breeds were generally not discerned. For chicken, we measured 30 wild RJF and 112 domestic chickens for the forelimb and 29 RJFs and 108 domestic chickens for the hind limb. In chicken, breed affiliation for many specimens was known (Table 1). Note that although most specimens were complete and we could sample measurements of their forelimbs as well as their hind limbs, not all forelimb and hind limb measurements were available for all specimens, resulting in some specimens being only represented by measurements of their forelimbs or their hind limbs, respectively (File S1). We did not consider the modern meat lines (broilers) due to their generally shortened life span, resulting in a degree of osteological maturation in most available specimens that could not be categorized as osteologically mature with the criterion used in this work, since the epiphyses of their long bones are not completely fused (*Zuidhof et al., 2014*; *Bennett et al., 2018*). We could not find skeletally mature broiler parent stock either in the collections visited.

We coded sternum characters in 63 specimens, from which 12 were RJF and the rest belonged to domestic breeds (seven Araucana, seven bantam, 16 Chabo, four Cochin,

three Italiener, one Kömpfer, four Polish, six Shamo, three Ukokkei). Note that it was not possible to code all characters in all specimens. We did not take measurements of the sterni because these bones are frequently broken in museum collections. Further, crooked carina sterni would have rendered linear measurements arbitrary and continuous ossification of the keel throughout life would have implemented ontogenetic variation.

The specimens used for this study are housed in the Paläontologisches Institut und Museum, Universität Zürich (Switzerland), Naturmuseum Senckenberg (Germany), Staatssammlung für Anthropologie und Paläoanatomie München (Germany), Zoologisches Institut/Populationsgenetik (former Institut für Haustierkunde), Christian-Albrechts-Universität zu Kiel (Germany), and The Natural History Museum at Tring, Bird Collection, and General and Darwin collection (United Kingdom). Raw measurements and information about the investigated specimens are provided in File S1.

### Analyses

We performed independent two-tailed student's $t$-tests to check for differences in the limb proportions in their respective limb between wild and domestic forms for all three species. Further, we performed independent two-tailed student's $t$-tests to check for differences in limb length, as a proxy for body size, between domestic and wild forms of the three species in this study.

In order to determine the explanatory power of body size on limb proportions in chicken, mallard ducks, and Muscovy ducks, we performed linear regressions using hind limb and forelimb total length as a proxy of body size, respectively. We contrasted the log-transformed values of the length of each bone with the log-transformed values of the total length of the correspondent limb. Limb length is the best body size approximation available, since the age of the used specimens might be variable, even though all were skeletally mature, and using the mean breed weight or size could be more problematic. To determine if the regression slopes were equal to 1 (null hypothesis of isometry), we determined if the 95% confidence intervals of the slopes included 1.0. We assumed positive or negative allometry to best explain a given scaling relationship if the confidence interval of the regression did not include 1, with a slope of >1.0 or <1.0, respectively. As a comparison and similar as for the adult sample described here, we extracted data on limb bone dimensions of chicken embryos and juveniles from the literature (*Thomas, Sadler & Cooper, 2016*; *Faux & Field, 2017*) and determined allometric growth patterns as described for the adult sample.

In the case of chicken, for which the breed affiliation of many specimens was known (Table 1), we also tested if there were significant differences in the limb proportions among breeds. For this, we used only domestic breeds with a sample size greater or equal to four to increase the rigor of our results. The breeds included in this analysis were thus Appenzeller Spitzhaubenhuhn, Araucana, bantam, Chabo, Cochin, Polish, Shamo, Ukokkei and white Leghorn, as well as the RJF. Due to small intra-breed sample sizes, we performed non-parametric Kruskal–Wallis tests, as well as post-hoc pairwise Wilcoxon rank sum tests to compare limb proportions among breeds.

Sternum characteristics were compared between wild and domestic chicken as well as among breeds of domestic chicken. For this, we compared percentages of character states in wild *vs.* domestic, as well as among the domestic breeds. For the latter comparison, we only used breeds for which the sample size was greater than four. The breeds included in this analysis were: Araucana, bantam, Chabo, Cochin, Ukokkei, Italiener, Kömpfer, Shamo and Polish, as well as the RJF. Due to small sample sizes, we refrained from statistically test dissimilarity of percentages.

## RESULTS

The pairwise comparisons of limb proportions in wild *vs.* domestic chicken, mallard duck, and Muscovy duck showed that significant differences in limb proportions only occured in the mallard duck (Fig. 4, Table 2). Domestic mallard ducks, compared to their wild counterpart, had proportionally longer humeri and shorter radii and carpometacarpi in the case of the wing, and proportionally longer tarsometatarsi and shorter femora in the case of the hind limb (Fig. 4, Table 2).

Pairwise comparisons of total limb length—as a proxy for body size—between wild and domestic forms in each species showed that the domestic varieties are larger than the wild ones in the mallard duck and the chicken, while we found no significant differences in the Muscovy duck (Fig. 5). In addition, variation of limb length appeared greater in domestic chicken and mallard ducks, compared to their wild counterparts (Fig. 5).

The analyses comparing the log transformed length of the limb bones with the log transformed total length of the limb—as a proxy for body size—showed that the lengths of the limb bones were strongly and significantly correlated with body size in all three species (Fig. 6, Table 3). We found different scaling patterns of limb bone lengths and body size (Table 3). Similar analyses featuring limb bone length *vs.* body size in ontogenetic samples extracted from *Thomas, Sadler & Cooper (2016)* and *Faux & Field (2017)* showed allometric growth in several of the bones during early chicken ontogeny (Fig. 7, Table 4).

Comparisons of limb proportions among chicken breeds revealed significant differences (Fig. 8, Table 5, File S2). A comparison of the mean total forelimb length of the different chicken breeds, showing their size differences, is provided in Table 6.

Our observations on the characters of the sternum show that for most characters, both character states could be found in wild as well as domestic chicken (Table 7). The only exception was that a spongy foramen pneumaticum and pars cardiaca were never reported in wild RJF (Table 7). Further, frequency of character states between wild and domestic were different regarding the carnina sterni, which tended to be more frequently non-notched in domestic chicken, while it tended to be more frequently notched in wild RJF (Table 7). Crooked carina sterni could be found in both, wild and domestic chicken (Table 7). Among the domestic chicken breeds, none of the character states were found to be exclusive for a particular breed (Table 7). However, we also found a hint to an unequal presence of characteristics among breeds (Table 7). Notably, bantam tended to straight carina sterni and fanned trabecula mediana more so than other breeds, and especially the chabo (Table 7). Further, we found that most breeds, except the chabo tended to exhibit a straight, rather than a crooked carina sterni.

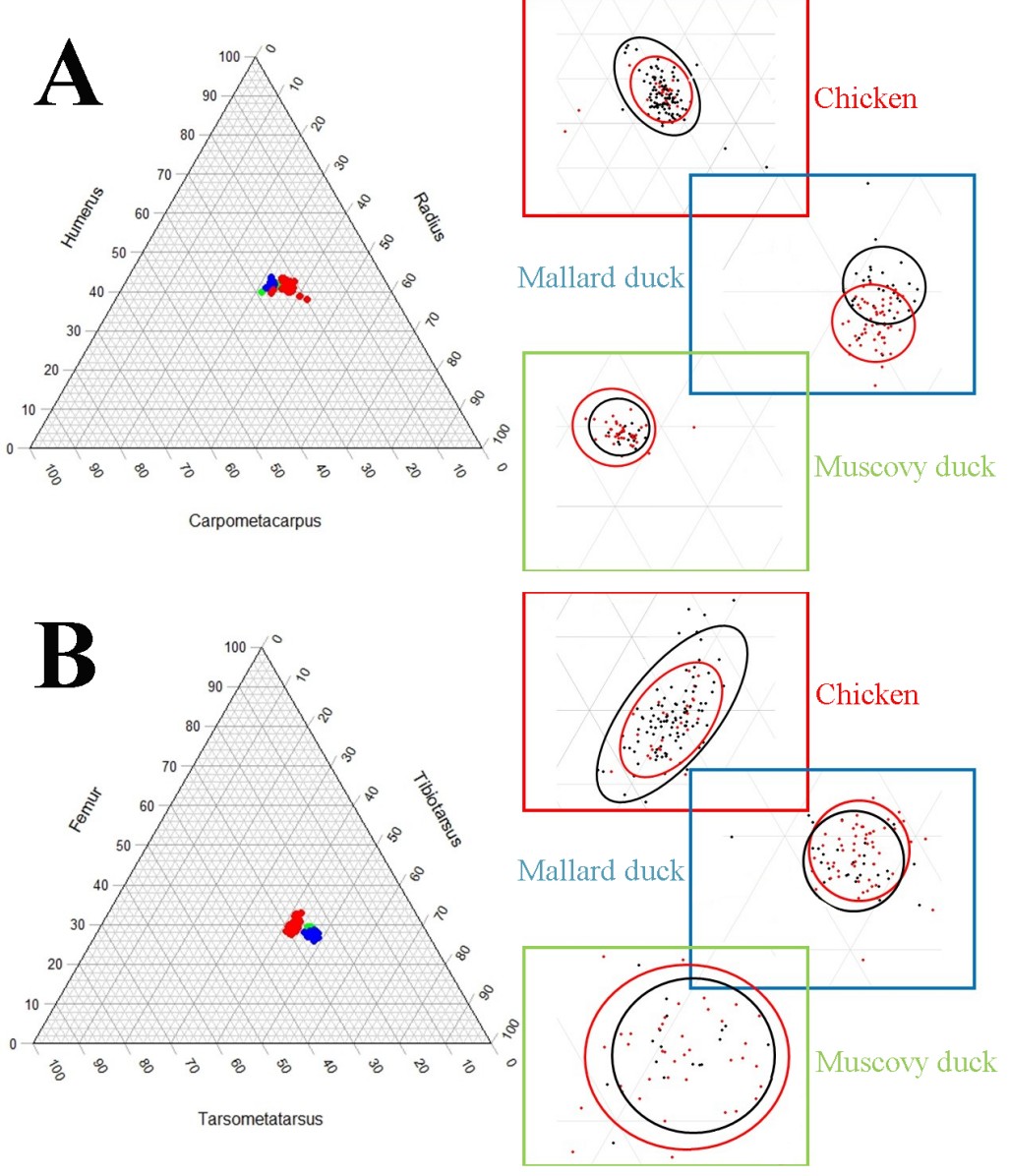

**Figure 4 Ternary plots illustrating the limb bone proportions in chicken (red datapoints), mallard duck (blue datapoints) and Muscovy duck (green datapoints) in the forelimb (A) and the hind limb (B).** Axes are representing the proportions of each bone in the limb as a percentage of the whole limb length (see Fig. 2). Each plot is accompanied with zoomed in areas corresponding to each species differentiating wild forms (red dots) and domesticated forms (black dots). Circles in zoomed areas capture where most points for wild or domestic forms fall. (For statistical comparison of proportions see Table 2).

## DISCUSSION

Our results show that, unlike wild and domestic forms of chicken and Muscovy ducks, domestic mallard ducks had a set of proportions in the limb bones that was significantly different from the wild form (Fig. 4, Table 2). At the same time, we observed that domestic

**Table 2** Results of two-tailed, independent $t$-test for the proportions of long bones of the forelimbs and hind limbs of domestic *vs.* wild forms of chicken, mallard duck, and Muscovy duck.

| Comparison | Df | T | *p*-value |
|---|---|---|---|
| **Chicken** | | | |
| Humerus | 70.893[*] | −0.746 | 0.4584 |
| Radius | 140 | 1.674 | 0.0963 |
| Carpometacarpus | 140 | −1.603 | 0.1112 |
| Femur | 133 | 1.477 | 0.1420 |
| Tibiotarsus | 135 | −1.111 | 0.2685 |
| Tarsometatarsus | 135 | −0.982 | 0.3279 |
| **Mallard duck** | | | |
| Humerus | 86 | 5.212 | **<0.0001** |
| Radius | 45.049[*] | −2.520 | **0.0154** |
| Carpometacarpus | 43.519[*] | −3.166 | **0.0028** |
| Femur | 86 | −1.989 | **0.0498** |
| Tibiotarsus | 45.171[*] | −1.001 | 0.3222 |
| Tarsometatarsus | 86 | 3.042 | **0.0031** |
| **Muscovy duck** | | | |
| Humerus | 54 | −0.199 | 0.8430 |
| Radius | 54 | 1.390 | 0.1701 |
| Carpometacarpus | 54 | −0.899 | 0.3728 |
| Femur | 54 | 0.730 | 0.4684 |
| Tibiotarsus | 54 | −0.408 | 0.6849 |
| Tarsometatarsus | 54 | −0.271 | 0.7873 |

**Notes.**
[*]The number of degrees of freedom is approximated by the Welch–Satterthwaite formula.
Bold: null hypothesis of equality of means rejected. Df, degrees of freedom.

mallard ducks were markedly larger than their wild relatives (Fig. 5). In cases where body size differences between wild and domestic forms are marked, differences in limb proportions between wild and domestic forms might be attributable to allometric scaling. Limb proportions were found to scale allometrically with body size in several limb bones in mallard ducks (Fig. 6, Table 3). The observed difference of limb proportions in wild and domestic mallard ducks (Fig. 4, Table 2) could thus be interpreted as the result of allometric scaling due to particularly pronounced size differences in the wild *vs.* the domestic sample. Specifically, in the case of the mallard duck's forelimb, the humerus scales positively allometrically, the radius isometrically, and the carpometacarpus negatively allometrically, leading to proportionally longer humeri and shorter radii in larger specimens (Figs. 2 and 4, Table 3).

Although no ontogenetic study is available for mallard ducks as a comparison, this same tendency for allometric scaling of the limb bones can be extracted from the data available in the literature regarding embryological and post-hatching growth of chicken (*Jackson & Diamond, 1996*; *Thomas, Sadler & Cooper, 2016*; *Faux & Field, 2017*) (Fig. 7). As the body size of the embryo and hatched birds increases during growth, their limb proportions change in a similar tendency (except tibiotarsus and tarsometatarsus in

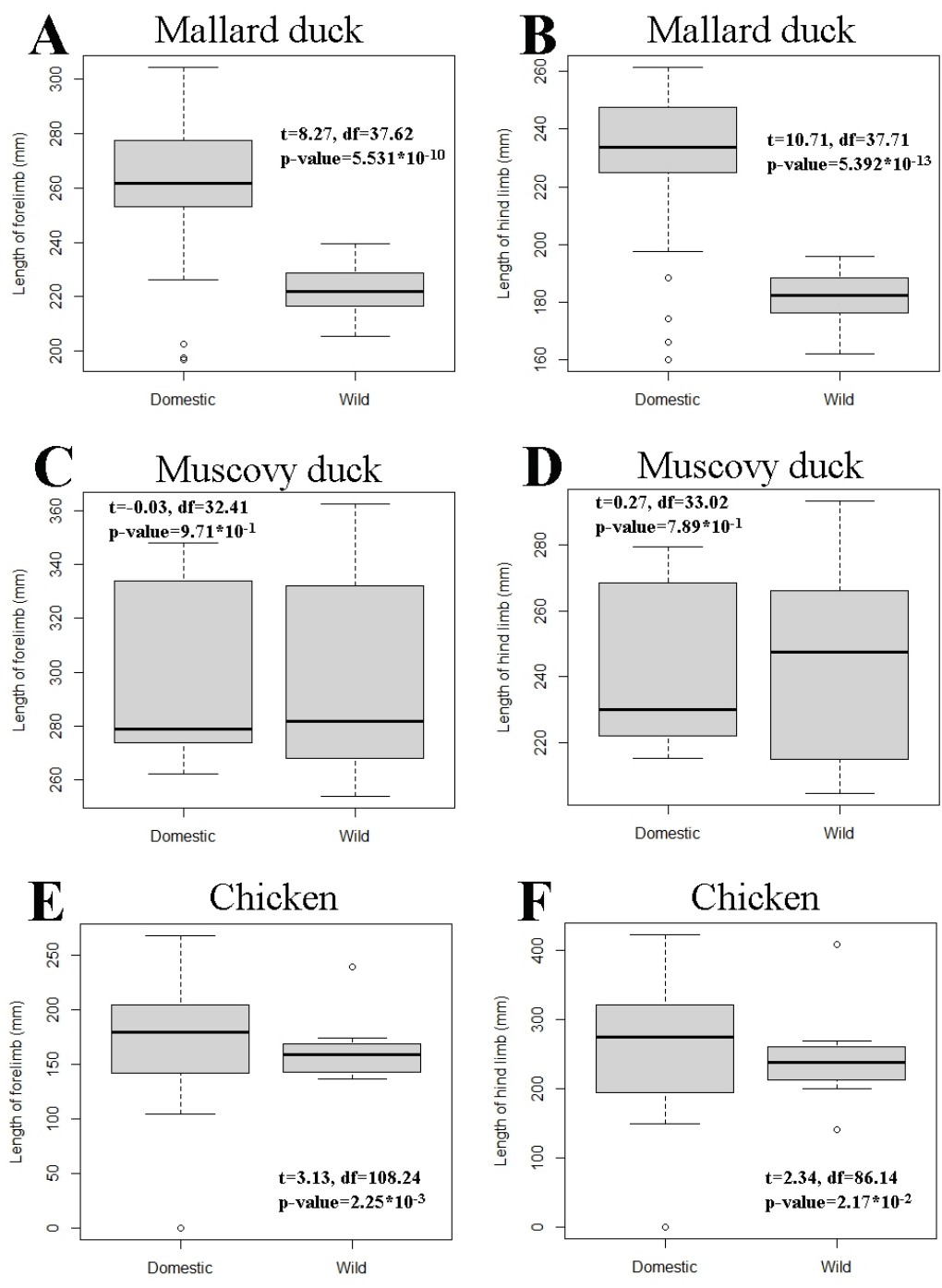

**Figure 5** **Comparison of limb length between wild and domestic forms in mallard duck (A and B), Muscovy duck (C and D), and chicken (E and F) in box plot form.** The box of the box plots contains the interquartile range going from the first quartile to the third quartile. The middle line represents the median. Results of *t*-tests are included and show t statistics (t) with associated degrees of freedom (df) and *p*-values.

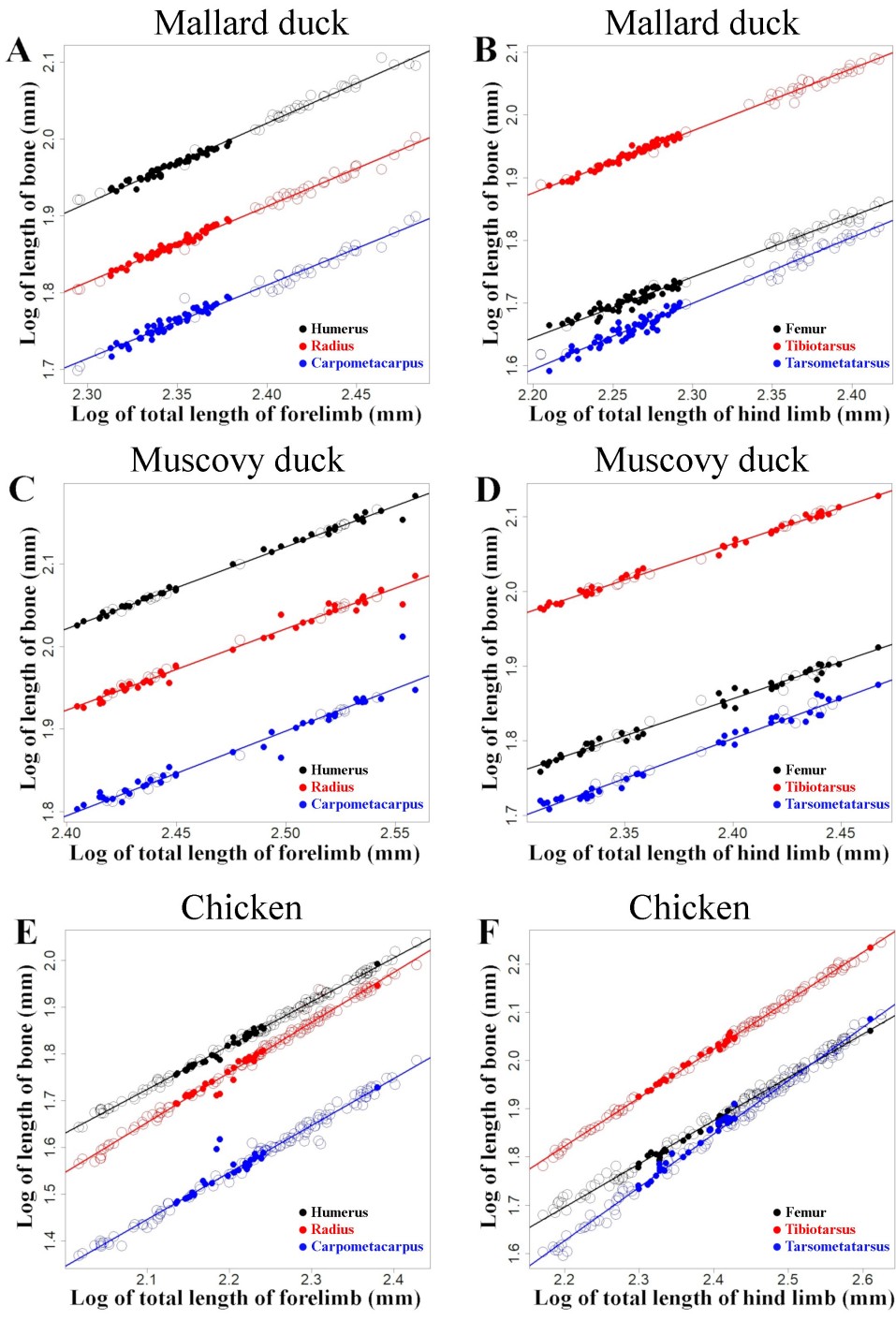

**Figure 6** **Regressions of log-transformed length of the studied limb bones *vs.* log-transformed lengths of the respective limb as a proxy for body size.** These regressions are shown for mallard ducks (A and B), Muscovy ducks (C and D), and chicken (E and F). Domestic forms are represented with open circles and wild forms with filled points. Colours denote humerus and femur (black), radius and tibiotarsus (red), and carpometapcarpus and tarsometatarsus (blue), respectively. Straight lines indicate least squares regression lines.

**Table 3 Results of linear regressions of long bone lengths *vs.* total limb length (*i.e.*, body size) in chicken, mallard ducks, and Muscovy ducks.**

| Comparison | a (ci) | Scaling pattern | $r^2$ | F (df) | *p*-value |
|---|---|---|---|---|---|
| **Chicken** | | | | | |
| Humerus | 0.937 (0.926, 0.948) | Negative allometry | 0.9954 | $3.002*10^4$ (1, 140) | $<2.2*10^{-16}$ |
| Radius | 1.069 (1.054, 1.085) | Positive allometry | 0.9923 | $1.809*10^4$ (1, 140) | $<2.2*10^{-16}$ |
| Carpometacarpus | 1.004 (0.981, 1.027) | Isometry | 0.9813 | $7.342*10^3$ (1, 140) | $<2.2*10^{-16}$ |
| Femur | 0.896 (0.881, 0.911) | Negative allometry | 0.9903 | $1.38*10^4$ (1, 135) | $<2.2*10^{-16}$ |
| Tibiotarsus | 1.005 (0.998, 1.013) | Isometry | 0.998 | $6.822*10^4$ (1, 135) | $<2.2*10^{-16}$ |
| Tarsometatarsus | 1.105 (1.088, 1.122) | Positive allometry | 0.9918 | $1.636*10^4$ (1, 135) | $<2.2*10^{-16}$ |
| **Mallard duck** | | | | | |
| Humerus | 1.038 (1.016, 1.061) | Positive allometry | 0.9898 | $8.336*10^3$ (1, 86) | $<2.2*10^{-16}$ |
| Radius | 0.985 (0.965, 1.006) | Isometry | 0.9907 | $9.137*10^3$ (1, 86) | $<2.2*10^{-16}$ |
| Carpometacarpus | 0.956 (0.926, 0.986) | Negative allometry | 0.9791 | $9.903*10^3$ (1, 86) | $<2.2*10^{-16}$ |
| Femur | 0.977 (0.941, 0.996) | Negative allometry | 0.9844 | $5.423*10^3$ (1, 86) | $<2.2*10^{-16}$ |
| Tibiotarsus | 0.99 (0.973, 1.008) | Isometry | 0.9933 | $1.268*10^4$ (1, 86) | $<2.2*10^{-16}$ |
| Tarsometatarsus | 1.054 (1.023, 1.085) | Positive allometry | 0.9819 | $4.677*10^3$ (1, 86) | $<2.2*10^{-16}$ |
| **Muscovy duck** | | | | | |
| Humerus | 0.992 (0.971, 1.013) | Isometry | 0.9941 | $9.336*10^3$ (1, 54) | $<2.2*10^{-16}$ |
| Radius | 0.987 (0.954, 1.021) | Isometry | 0.9849 | $3.513*10^3$ (1, 54) | $<2.2*10^{-16}$ |
| Carpometacarpus | 1.026 (0.968, 1.083) | Isometry | 0.9595 | $1.28*10^3$ (1, 54) | $<2.2*10^{-16}$ |
| Femur | 0.988 (0.954, 1.022) | Isometry | 0.9846 | $3.455*10^3$ (1, 54) | $<2.2*10^{-16}$ |
| Tibiotarsus | 0.97 (0.95, 0.99) | Negative allometry | 0.9944 | $9.641*10^3$ (1, 54) | $<2.2*10^{-16}$ |
| Tarsometatarsus | 1.068 (1.029, 1.108) | Positive allometry | 0.9819 | $2.933*10^3$ (1, 54) | $<2.2*10^{-16}$ |

**Notes.**

a, regression slope with corresponding 95% confidence interval (ci), scaling pattern, interpretation of isometry (ci including 1, $a = 1$), negative allometry (ci not including 1, $a < 1$)), or positive allometry (ci not including 1, a>1)); r2, coefficient of determination; F, F statistic with associated degrees of freedom (df). For visualizations see Fig. 6.

post-hatching growth, Fig. 7, Table 4) as we observe in differently sized adult chickens (Fig. 6, Table 3). Similar scaling relationships were also reported for wing growth in *Larus californicus*, which shows a progressive decrease in the humerus length relative to the radius and carpometacarpus (*Middleton & Gatesy, 2000*). This suggests that the differences in proportions of the limbs among adult mallard ducks (static allometry) might be at least partially related to allometric scaling during development and growth (ontogenetic allometry). Although allometric scaling of limb bones was also observed in chicken and Muscovy ducks (Fig. 6, Table 3), the lack of pronounced size differences between wild and domestic forms in these species (Fig. 5) could explain the observed lack of proportional differences of limb bones in the wild *vs.* domestic forms in these species (Fig. 4, Table 2).

Different allometric scaling patterns of limbs bones probably have functional implications. For example, it has been suggested that there is an inverse correlation of the proportion of the humerus in the wing with aerial maneuverability in birds (*Middleton & Gatesy, 2000*). Moreover, flightlessness is reportedly linked to a proportionally long humerus compared to the rest of the long bones of the wing in theropods (*Middleton & Gatesy, 2000*) and steamer ducks (*Tachyeres*) (*Livezey & Humphrey, 1986*).

Experiments on the domestication of the wild mallard duck resulted in bigger body sizes and rapid loss (after three generations) of the ability to fly (*Darwin, 1868*). *Mason*

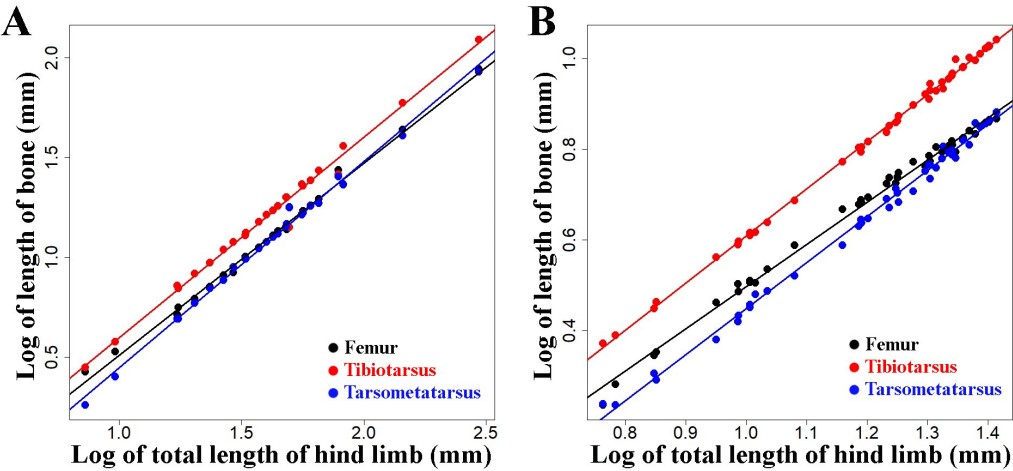

**Figure 7** Regressions of log-transformed length of hind limb bones *vs.* log-transformed lengths of the hind limbs as a proxy for body size in embryonic and juvenile chicken. These egressions depict scaling relationships for 7–17 days old chicken embryos (A, data taken from *Faux & Field, 2017*) and in 1–36 weeks old post-hatching chicken chicks (B, data taken from *Thomas, Sadler & Cooper, 2016*). Colours denote femur (black), tibiotarsus (red), and tarsometatarsus (blue), respectively. Straight lines indicate least squares regression lines.

**Table 4** Results of linear regressions of long b one lengths *vs.* total limb length of chick en embryos (*Faux & Field, 2017*) and juveniles (*Thomas, Sadler & Cooper, 2016*).

| Comparison | a (ci) | Scaling pattern | $r^2$ | F (df) | p |
|---|---|---|---|---|---|
| *Faux & Field (2017)* | | | | | |
| Femur | 0.963 (0.928, 0.998) | Negative allometry | 0.9628 | $3.242{*}10^3$ (1, 23) | $<2.2{*}10^{-16}$ |
| Tibiotarsus | 1.005 (0.958, 1.051) | Isometry | 0.9886 | $1.993{*}10^3$ (1, 23) | $<2.2{*}10^{-16}$ |
| Tarsometatarsus | 1.033 (1.001, 1.065) | Positive allometry | 0.9948 | $4,403{*}10^3$ (1, 23) | $<2.2{*}10^{-16}$ |
| *Thomas, Sadler & Cooper (2016)* | | | | | |
| Femur | 0.930 (0.909, 0.952) | Negative allometry | 0.9943 | $7.549{*}10^3$ (1, 43) | $<2.2{*}10^{-16}$ |
| Tibiotarsus | 1.040 (1.027, 1.053) | Positive allometry | 0.9983 | $2.5{*}10^4$ (1,43) | $<2.2{*}10^{-16}$ |
| Tarsometatarsus | 1.016 (0.994, 1.038) | Isometry | 0.9951 | $8.788{*}10^3$ (1,43) | $<2.2{*}10^{-16}$ |

**Notes.**
a, regression slope with corresponding 95% confidence interval (ci); scaling pattern, interpretation of isometry (ci including 1, $a = 1$), negative allometry (ci not including 1, a < 1)), or positive allometry (ci not including 1, a > 1)); r2, coefficient of determination; F, F statistic with associated degrees of freedom (df); p = p-value. For visualizations see Fig. 7.

*(1984)* proposed that this loss of flight capacity might be due to the loss of positive selection for flying ability: under the care of humans, mallard ducks are usually granted access to reliable sources of food at specific locations, thus flight becomes less essential for survival. The loss of the capacity to fly may thus be related to the proportion of the limbs, which upon less selection due to lack of necessity to fly may have become significantly different from their wild counterpart. An alternative explanation could be that the selection of the domesticated mallard ducks for specific traits could have pleiotropic effects that lead to a reduction of flight capacity. In certain fancy breeds, limb proportions might even be under directed artificial selection. These hypotheses remain to be tested in future works.

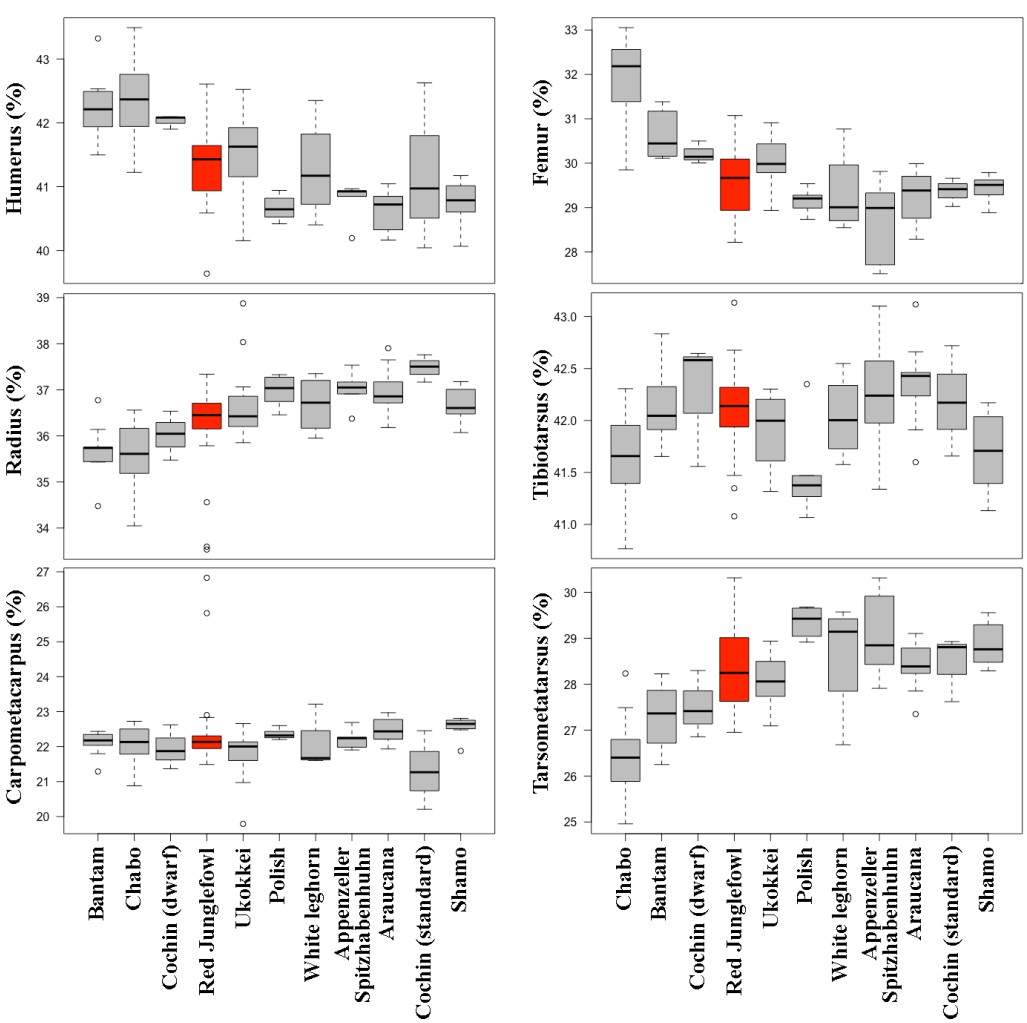

**Figure 8 Comparison of limb bone proportions among the here investigated chicken breeds in box plot form.** Each graph depicts the proportion of one limb bone as a percentage of the respective limb's length (forelimb bones shown on the left panels, hind limb bones on the right panels). Only breeds with a sample size ≥ 4 were included. Cochin is divided in its dwarf and standard forms. Data on the wild relative of domestic chicken (red junglefowl) are provided as comparison and highlighted in red. The box of the box plots contains the interquartile range going from the first quartile to the third quartile. The middle line represents the median. Boxes are ordered from smaller to bigger total length of the limb for the group (values of these lengths in Table 6).

Other skeletal, muscular and perhaps neurological changes may also be related to the loss of flight capacity. In the case of the hind limb, we observed in domestic forms of the mallard duck a relative shortening of the femur and elongation of the tarsometatarsus. *Gatesy & Middleton (1997)* reported an apparent gradient of cursoriality increasing with tarsometatarsal proportion among theropod species with similar tibiotarsus/femur ratios. The pattern we observed in domestic mallard ducks, together with a decrease in flight adaptations (above), is consistent with a more terrestrial, although not necessarily highly cursorial, lifestyle of the domesticated *vs.* the wild form.
**Table 5** Results of Kruskal–Wallis tests for comparison of limb proportions among chicken breeds.

| Skeletal element | Chi-squared | df | *p*-value |
|---|---|---|---|
| **Forelimb** | | | |
| Humerus | 65.52 | 8 | 3.81E−11 |
| Radius | 46.3 | 8 | 2.09E−07 |
| Carpometacarpus | 21.52 | 8 | 0.005894 |
| **Hind limb** | | | |
| Femur | 61.37 | 8 | 2.51E−10 |
| Tibiotarsus | 31.01 | 8 | 1.4E−04 |
| Tarsometatarsus | 58.41 | 8 | 9.57E−10 |

**Table 6** Mean length of the forelimb and hind limb of the investigated chicken breeds. Only breeds with a sample size ≥ 4 were included. Data on the wild relative of domestic chicken (red junglefowl) are provided as a comparison.

| Form/breed | Forelimb length (mm) | Hind limb length (mm) |
|---|---|---|
| Chabo | 123.57 | 172.94 |
| Bantam | 116.11 | 175.11 |
| Cochin (dwarf) | 130.72 | 194.94 |
| RJF | 158.83 | 241.44 |
| Ukokkei | 168.93 | 254.19 |
| Polish | 188.86 | 293.25 |
| White leghorn | 192.52 | 297.66 |
| Appenzeller Spitzhaubenhuhn | 197.03 | 307.31 |
| Araucana | 202.10 | 312.80 |
| Cochin (standard) | 227.97 | 359.18 |
| Shamo | 231.76 | 365.59 |

**Table 7** Characters and character states of the sternum in the investigated domestic chicken breeds and the red junglefowl. Numbers indicate the percentage of specimens in that form/breed exhibiting the respective character state. Numbers in brackets indicate the total number of specimens per character state. Note that it was not possible to code all characters in all specimens and that for single breed comparisons only breeds with n ≥ 5 are shown. The group 'domestic' summarises seven Araucana, seven Bantam, 16 Chabo, four Cochin, three Italiener, one Kömpfer, four Polish, six Shamo, and three Ukokkei specimens. For detailed descriptions of characters and states see text and Fig. 3.

| Form/breed (n) | Carina sterni | | Carina sterni | | PC and FP | | Trabecula mediana | |
|---|---|---|---|---|---|---|---|---|
| | Crooked | Straight | Notched | Non-notched | Spongy | Smooth | Fanned | Straight |
| Araucana (7) | 0.14 (1) | 0.86 (6) | 0.14 (1) | 0.86 (6) | 0.57 (4) | 0.43 (3) | 0.60 (3) | 0.40 (2) |
| Bantam (7) | 0.00 (0) | 1.00 (7) | 0.29 (2) | 0.71 (5) | 0.29 (2) | 0.71 (5) | 1.00 (7) | 0.00 (0) |
| Chabo (16) | 0.50 (8) | 0.50 (8) | 0.19 (3) | 0.81 (13) | 0.38 (6) | 0.63 (10) | 0.69 (11) | 0.31 (5) |
| Shamo (6) | 0.33 (2) | 0.67 (4) | 0.33 (2) | 0.67 (4) | 0.33 (2) | 0.67 (4) | 0.40 (2) | 0.60 (3) |
| Domestic (51) | 0.39 (20) | 0.61 (31) | 0.20 (10) | 0.80 (41) | 0.39 (20) | 0.61 (31) | 0.63 (30) | 0.38 (18) |
| Wild/RJF (12) | 0.50 (6) | 0.50 (6) | 0.75 (9) | 0.25 (3) | 0.00 (0) | 1.00 (12) | 0.83 (10) | 0.17 (2) |

**Notes.**

FP, foramen pneumaticum; n, total number of specimens; PC, pars cardiaca; RJF, red junglefowl.
Changes in limb proportions were not observed in the Muscovy duck. Its limb proportions remained indistinguishable from the wild form, as opposed to the mallard duck (Fig. 4, Table 2). Changes in flight habits might not have been similar between the wild and domestic forms of mallard and Muscovy ducks While some populations of wild mallard ducks are migratory (*Bellerose & Crompton, 1970*; *Bird Life International, 2019*), their domestic variants rarely fly as much and many breeds lack flight capacity entirely (*Smith, Pederson & Kaminski, 1989*). Domestic Muscovy ducks, on the other hand, although non-migratory birds (*Accordi & Barcellos, 2006*), have been reported to be strong flyers (*Swatland, 1980*), suggesting that their flight habits might not have changed so markedly in domestication as to produce a different morphology. Alternatively, and related to the allometric scaling relationships considered above, it is noteworthy that in contrast to the other two species, limb bones of the forelimb were found to consistently scale isometrically in the Muscovy duck and body sizes were similar in the wild and the domestic groups (Fig. 5, Table 3), suggesting restricted evolvability concerning forelimb proportions in this species. An analogous conclusion was reached when postulating lower evolvability of facial shape in cats *vs.* dogs when reporting largely isometric *vs.* allometric growth in their skulls, respectively (*Sánchez-Villagra et al., 2017*). Further, timing of domestication might influence the differences which we see in the investigated species. While it was estimated that the domestication process has started about 3,500 years before present in the chicken (*Peters et al., 2016*) and about 2,200 years ago in the mallard duck (*Zhang et al., 2018*), much less is known about the temporal framework of Muscovy duck domestication. Minimum estimates for the onset of Muscovy duck domestication range between 600–700 years before present (*Mason, 1984*). Although most mammal domesticates have been estimated to start into this process much earlier (*e.g.*, maybe more than 12,000 years in domestic dogs; *Larson & Fuller, 2014*), significant limb proportional differences in domestic *vs.* wild mallard ducks as well as among domestic chicken breeds suggest that the relatively short timeframe of domestication in the investigated bird species has been sufficient for such changes to occur. In contrast, the Muscovy duck's probably more recent domestication history might be too short a timeframe as to lead to extensive changes of limb proportions.

Although we observed no significant differences in the proportional length of the limb bones in wild *vs.* domestic chicken (Table 2), examination of separate breeds revealed significant inter-breed differences in terms of size and limb proportions (Table 6, Fig. 8, Table S1). One possible interpretation is for diversification to have taken multiple directions in the case of the limb proportions in domestic birds, but selection has not provoked a unique directionality, maintaining the same mean proportions between domestic and wild forms in all bones, and therefore obscuring the variation that can only be observed when individual breeds are compared. The most notable case of breed-specific specialization is the chabo, also known as Japanese bantam. Chabo is a true bantam breed (*i.e.*, it does not present a large variant) with characteristically short legs (*Roberts, 2008*). The short leg trait is associated with a dominant allele of gene *Cp*, which is lethal in homozygosis (*Shibuya, Fujio & Kondo, 1972*; *Shibuya & Kuroda, 1973*) and is selected as a distinct characteristic of the breed (*Van Grouw, 2018*). The *Cp* gene accelerates differentiation of chondrocytes accompanied by suppression of their proliferation, leading to stunted limb long bone

growth (*Shibuya, Fujio & Kondo, 1972*). Similar phenotypes are also known in some breeds of domestic dogs (*e.g.*, dachshund, corgi) and cats, and represent cases of specialization in breed formation (*Rimbault & Ostrander, 2012*). As in the case of the mallard duck, the differences in adult limb proportions in chicken might be due to allometric scaling during development and growth (see above). Differences in limb proportions among breeds could thus be related to the different sizes of breeds. This is consistent with the observation of allometric scaling of the bones depending on size in chicken (Fig. 6 and Table 3).

Our observations on the characters of the sternum show that the only feature that appears to be associated with domestication in chicken is sponginess of the foramen pneumaticum and the pars cardiaca (Table 7). Further, the notch in the carina sterni seems to be a rare condition in domestic chicken, while it is the most common condition in the wild form (Table 7). Underlying reasons for both differentiations of character states between wild and domestic may be related to the functionality of the sternum, which is likely reduced in most domesticates compared to the wild state due to a tendency to flightlessness in the first, or to body size changes related to domestication (see above). The RJF develops a crooked keel in half of the observed cases (Table 7). The crookedness of the keel has been shown to be hereditary and influenced by early roosting conditions (*Blount, 1933*; *Warren, 1937*; *Waters, 1949*). Our observations reveal that this is not a feature unique to domestic varieties, as has been previously reported (*Kittelsen et al., 2021*).

A majority of domestic variants exhibited straight carina sterni, a trait that is even found in all of the studied bantam specimens (Table 7). This is with the notable exception of the chabo, in which half the investigated specimens exhibited a crooked carina sterni (Table 7). A possible explanation might be that the shortness of the legs in the chabo is responsible for a more direct contact with higher pressure on the keel with the perching structure of their environment, as it has been shown to be a determinant factor for its appearance (*Blount, 1933*; *Warren, 1937*; *Waters, 1949*; *Pickel, Schrader & Scholz, 2011*; *Casey-Trott et al., 2015*). Behaviour, especially flying and perching, could be an important factor distinguishing breeds but could not be assessed in this study. In summary, these results reveal that the variation of the different characters are not segregated among the breeds.

## CONCLUSIONS

This study shows how the allometric growth of birds can help us understand the variation in limb proportions of differently sized domestic breeds. Furthermore, we show that significant differences in the proportions of limb bones of wild and domestic mallard ducks exist, whereas no significant differences between wild and domestic forms of chicken and Muscovy ducks were found. However, when individual chicken breeds were analysed, differences between breeds could be observed. We offer possible explanations for these differences such as an extension of the allometric growth of domestic mallard ducks, loss of positive selection for flying ability, and/or pleiotropic effects or evolvability constraints, which could inspire future research. Furthermore, we observed that the condition of crooked keel is not unique to domestic chicken, that some sternal characteristics are more frequent in certain breeds, but that overall there were no keel characteristics that are unique

for certain breeds. These findings add new observations to the study of keel characteristics that can be added to the body of knowledge of domestication and poultry welfare. In sum, this study highlights the lack of universal patterns in domestication and breed formation.

## ACKNOWLEDGEMENTS

We thank Jorge Carrillo-Briceño, Gabriel Aguirre-Fernández, and Evelyn Hüppi (Universität Zürich, UZH) and Alicia Batuecas Suárez and Arturo Morales Muñiz (Universidad Autónoma de Madrid) for help with the preparation of the UZH materials, Mark Adams, Joanne Cooper and Hein van Grouw (The Natural History Museum, Bird Collection, Tring, General and Darwin collection), Henriette Obermaier (Staatssammlung für Anthropologie und Paläoanatomie München), Gerald Mayr (Naturmuseum Senckenberg), and Renate Lücht (Zoologisches Institut/Populationsgenetik, Christian-Albrechts-Universität zu Kiel) for granting access to the skeletal materials. We thank Judith Recht for editorial help, Erin Maxwell for discussion, John Hutchinson for editorial work, and Ida Thøfner and one anonymous reviewer for a constructive review that helped to improve the manuscript.

### Funding

This research was financially supported by the Stiftung für wissenschaftliche Forschung an der Universität Zürich (STWF-18-025), the Swiss National Science Foundation grant (31003A_169395 to Marcelo R. Sánchez-Villagra) and by the Federal Commission for Scholarships for Foreign Students (FCS, Switzerland to Daniel Núñez-León). The funders had no role in study design, data collection and analysis, decision to publish, or preparation of the manuscript.

### Grant Disclosures

The following grant information was disclosed by the authors:
Stiftung für wissenschaftliche Forschung an der Universität Zürich: STWF-18-025.
Swiss National Science Foundation: 31003A_169395.
Federal Commission for Scholarships.

### Competing Interests

The authors declare there are no competing interests.

### Author Contributions

- Carlos Manuel Herrera-Castillo conceived and designed the experiments, performed the experiments, analyzed the data, prepared figures and/or tables, authored or reviewed drafts of the paper, and approved the final draft.
- Madeleine Geiger, Daniel Núñez-León, Hiroshi Nagashima, Sabine Gebhardt-Henrich, Michael Toscano and Marcelo R. Sanchez-Villagra conceived and designed the experiments, authored or reviewed drafts of the paper, and approved the final draft.

## Data Availability

The raw measurements are available in the Supplementary File.

## Supplemental Information

Supplemental information for this article can be found online at http://dx.doi.org/10.7717/peerj.13229#supplemental-information.

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
