# Peer review of "Skeletal variation in bird domestication: limb proportions and sternum in chicken, with comparisons to mallard ducks and Muscovy ducks"

_PeerJ, doi:10.7717/peerj.13229_

## Round 0.1 · original submission · Major Revisions

Two reviewers have provided constructive critiques that will improve the paper. In addition to further proofreading and stylistic changes + restructuring, please pay heed to their points such as looking more closely at individual domestic breeds, and the analysis of sternal keel morphology. Be sure you address all points raised individually in your Response, which will aid re-review. Thank you for your interesting contribution.

Reviewer 1 ·

Basic reporting

overall the manuscript would benefit from proof readings to improve the general standard of English and sentence structure throughout, there are spelling errors that need correcting as well

I would like to see a broader introduction into the functional significance and importance of these areas of avian anatomy and morphology - there is no discussion of the role of the sternum in respiration that needs addressing for example - but a general broadening of the literature included in the introduction would better highlight the significance of this study - specific comments are detailed below.

tables could be improved as detailed - a numb hypothesis is stated but clearer aims and objectives would help

Experimental design

by incorporating the extra elements suggested in the introduction this will improve the definition of the research questions and better highlight how it fills a gap.

methods are ok but need to include more complete information as detailed 0 the sample sizes don't always seem to match up and swapping between common and species names is confusing at times

Validity of the findings

need clarification on comparisons made

Additional comments

overall the English could do with proof reading and corrections to sentence structure and spellings - too many sentence start with 'this' for example which is often non specific

line 35 correct mallard spelling

line 38 - this is unclear less intensive selection in which forms ?

line 52 - formatting error

line 53 use correct referencing format you don't need to give the title of the reference - its unclear if this is supposed to be a direct quote or not

introduction - this needs broadening to include a discussion of the role of the sternum in respiration

lines 99-100 - the language needs tightening in here, its unclear when you jump between common and breed strain names. what is the difference between fowl and chicken ? you need to specify what you are referring to here

line 116 - give model details and company name, country

line 119 - total length of the forelimb will depend on articulation of these bones not their overall length being summed it would db rare for the limb to be held completely straight - why was this parameter chosen for investigation ?

line 122 - be specific proportions of what?

line 123 - can you detail how the data from these groups were pooled or not pooled as the case may be

line 132 - why the variation in fore and hind limb numbers ?

line 132-141 - its hard to tally these numbers with the sample sizes that appear in the tables

line 194 - you state here that no differences were found in gallus but then on line 205 you choose to highlight 'the substantial size variation' surely they either are or are not different ?

line 208 -line 212 - jumping between common and species names is confusing do one or the other but not both

line 216 - again reference to the role in respiration would be appropriate here

line 219 - 222 - how can it be that you observe no differences but then when analysed individually you find differences - do you mean within and between groups ? needs clarifying what you analysed here

line 262 - this paragraph is confusing - the lack of a size difference is due to their being a lack of proportional differences - isn't this just trying to explain the lack of a difference by there being a lack of difference ? a functional explanation would improve this section

general comment - I think the difficulty in interpreting the data stem from the pooling of samples into wild and then everything else into domestic when clearly there are large variation in the domesticated breeds likely obscuring meaningful results

line 289 - 291- jumps between common and species names again

line 296 - are you referring to muscovy here ? 'this suggesting..' needs rewriting

line 300 how is this an analogous conclusion ?

line 300- linking or placing these changes in context of respiration needs to be done how would the sternum get damaged by resting when the muscles covers the bone ?

tables:

- adding sample size to table 1, table 2 should make clear if these samples are from the same Bird or different birds - it isn't clear what you mean by specimens and how these were pooled.

Table 2 why are some numbers underlined ? and what are there differences in fore and hind limbs measured

table 3 more information is needed in the legend

table 4 - the spacing of the table makes it hard to follow - are the r2 numbers spread over two lines ?

table 4 - the numbers and value don't need underlining if you aren't referring to anything specific

table 5 - nothing is bold in the table and if the names are in the table you don't need to specifically highlight RJF anyway - why do you only present these raw data ?

table 5 - why are numbers underlined ? why are the samples sizes different as you go across the table ?

figure 1 arrows not dots would be better and more clear, a fuller legends needs to explain what the figure is showing

figure 2 - p-value is written on the figure so it doesn't need explaining, im not sure the small images are helpful

·

Basic reporting

Please find comments to the manuscript entitled “Skeletal variation in bird domestication: limb proportions and sternum in chicken, with comparisons to mallard ducks and Muscovy ducks” below.

The submitted manuscript is generally well written. The impact of domestication on the biometric measures is an interesting topic. In the introduction a brief mentioning of when the species subjected to this study was domesticated compared to other species (e.g. mammals) would provide evidence of what to expect in terms of skeletal evolution. Is the timeframe of domestication too short to change the skeletal properties in the included species?
In both the introduction and discussion, reference to flying ability of Muscovy is referenced by blog post and unrefereed web pages, that is not considered as appropriate referencing and may if extensive weaken the study.

However, the manuscript suffers from being a bit unstructured in particular in the results sections. Here the lack of structure is hampering the interpretation and the readability of the results, due faulty references to figures and tables, to little detail in the tables for getting an overview of the wild and domesticated birds just to give some examples. This section need a thorough and careful revision with a clear focus on structure and logic order in the presentation of the results. See also specific comments below (the list is not exhaustive). More words/description on where there are differences will be an improvement and help the reader.
By aligning the structure of Materials and methods section and the Results section the readers ability to interpret the results presents will be considerably increased. A suggestion could be 1) wild-domesticated all species, followed by 2) chicken related comparisons, including breed differences for both leg and keel bones.

There has been a mixup in updating figure and table numbers in many of the cross references. That should be avoided, as it is hampering the interpretation of the results and evaluation of the conclusions.

Experimental design

In general the design of the study is sound, however validation of the measurement accuracy and instrument uncertainty was not addressed. Details on this is described below in specific comments.

As the breeding of poultry is an ongoing event, it will be useful to include the year/date where the specimens have been collected and included in their respective collection. Additionally inclusion of a sufficient number of birds from modern commercial lines, both egg and meat type birds would have been very useful for displaying the potential impact of intensive breeding selection where selection criteria is meat and egg yield.

Validity of the findings

The analysis of the limb measures is found to be valid and sound. However, the analysis of the keel bone data is next to absent. This is a major weak point and is elaborated several places below in both general and specific comments

Additional comments

When specific attention was given to the keel bone of the chicken together with reference to keel bone damage (both fractures and deformities) it is hard to understand why specific measurements of various keel bone dimensions (e.g. length and height of the carina, total length, distances between distal/caudal ends of trabeculae etc.) was not performed and compared to limb properties and also between wild and domesticated birds. This information is next to necessary for the understanding of keel bone fractures in layer hens.
As described above the lack of detailed morphometric measures of keel bones, which could have been done at the same time as the initial registration, is weakening the results. It should be included, including comparative analysis to the limp data. If not possible, all sections regarding the keel should be taken out of the manuscript. At its present form it does contribute significantly to the rest of the study.
Furthermore inclusion of modern adult commercial layers and meat type broiler breeders would greatly increase the understanding of the selective pressure that these two types of chicken have been put through the last 3-5 decades.

Specific comments:
Line 52: At the start of the line, there is an empty gap. It is unclear whether some text/wording is missing or if it is merely a technical issue.
Line 71-73: References to the muscovy ducks’ flight capacity may benefit considerably by using scientific and/or textbook literature or other refereed information instead of using anecdotal information from internet blog posts. For example a comparison of allometrics of skeletal bones and potential ability to fly in muscovy and pekin ducks by Swatland (1980). In the paper by Converse and Kidd (2001), the muscovy duck is considered a migratory bird. Poultry breeding standard reference may also be able to provide information on ability to fly.
Line 82-88: Crookedness of keel was also described by Buckner et al. (1949).
Line 88-90: The impact of perches on keel bone deformities in laying hens was recently reviewed by the EFSA panel (EFSA, 2015). Please clarify that this relations is not purely historical.
Line 93-94: The proposed definition of a keel bone fracture seem somewhat imprecise. The definition of a fracture in medical terms is referring to a discontinuation (breaking) of the bone tissue resulting a variety of different fracture manifestations where the involved bone is completely or incompletely disrupted with or without fragments. Fracture definition and characteristics are well described in a text book chapter by Marchiori (2013). A fracture will depending on the level of post fracture immobility and will heal more or less deformed with or without medical intervention. If a fracture is left unstable/insufficiently immobilized the bone repair tissue (callus) with “grow” excessively and new bone formation will appear close to the fracture line and “on top” of the bone surface as exostoses. This callus may in extreme cases be palpable for even the inexperienced after receiving a brief training. Callus and exostoses are always visible on x-ray. Please review the fracture definition accordingly. References to keel bone fracture characterization/visualization are to some extend available using different diagnostic modalities (e.g. imaging, histology, pathology) (Eusemann et al., 2018; Fleming et al., 2004; Richards et al., 2011; Rufener et al., 2018; Scholz et al., 2008; Thøfner et al., 2020; Tracy et al., 2019).
Line 115-116: To overcome the risk of applying unknown degrees of inaccuracy and uncertainty of the measurements and measuring equipment both replicate measurements and calibration of equipment is of great value. At present it is unclear whether a bone was measures several times (technical replicates). Please specify, and justify if not more than one measurement. What were the approaches/procedures for calibration of the calipers?
Line 119-121 : the total length of given limb “was taken”, “was defined” seems more appropriate as the tarsus/carpus and toes/phalanges were not included.
Line 122-123 (Table 1): The distribution of wild and domesticated birds is unclear for all three species. Please clarify. The pooling of all the wild and domesticated birds within each species is disturbing, especially in chickens, where many different breeds in all sizes are included. Please justify the reasoning. It would increase the immediate readability if table 1 was presented as an overview of the distribution and numbers of wild and domesticated birds within each species and note the breeds of mallard ducks and chickens, maybe including the mean length of the specific bones.
Line 133-136: It is acknowledged that juvenile modern meat type chickens were not investigated. Nevertheless, the study will benefit greatly by including adult broiler breeder birds, as these chicken line have been bred intensely for meat development. This has significantly altered their gait to a level where it is considered the new “normal” that broilers have increased gait scores compares to other breeds/line including commercial layer type hens, like the leghorn-derived lines.
Line 153 (table 3): Table 3 is referenced the manuscript, but the information provided in the manuscript in table 2. Again, presentation of the result can be improved. The means/medians and the variation of each breed would help the reader to interpret where the differences between breeds may be present. Please consider illustrating by using a plot/figure instead.
Line 179-191: The absence of specific measurements of keel dimensions is weakening the results considerably with regards to the keel morphology and its potential importance to keel bone fracture development observed in in particular highly productive laying hens.
Line 197: reference to table 2 is wrong. The table 2 provided in the manuscript is on the different chicken breeds included in the study. Maybe it is the content of table 1 that is referred to. Including presentation of the mean properties for reference will greatly improve the presentation of the results. Maybe move up fig 8 for this part…
Line 202: Table 3 is referenced the manuscript, but the information provided in the manuscript in table 2 is presenting the results described. As mentioned above (line 153).
Line 205-212: The lack of structure in the whole results section together with the faulty reference to figures and table numbering is impairing the review of this last paragraph too much.
Line 221-223: By including origin of breed formation, overall type of chicken (layer/meat, large/giant/dwarf size, height/weight span) in the table with the number of bird within the specific chicken breeds would substantiate the interpretation and discussion of the breed differences among the chicken breeds including RFJ. This can also be used for discussion the selective pressure of certain traits. In relation to that the absence of modern highly selected commercial lines, both layer and meat type chickens is weakening the conclusions. The selective pressure of the commercial lines may be considered the highest and fastest. Maybe visualized by creation of a small phylogentic tree including wild and domestic forms as well as breeds. Interactive Tree Of Life is an online tool for the display, annotation and management of phylogenetic and other trees (https://itol.embl.de/gallery.cgi) (Letunic and Bork, 2019).
Line 235-249 and fig 3 and 8: Some attention to the large variation in limp and bone lengths in both domesticated chicken and mallard breeds should be given. In particular as many of the different breeds have been bred for exterior traits including (body) size, therefore also known as “fancy breeds”. The observed differences may be partially explained by this. As mentioned above an overview of the standard size/dimensions, age of breed, intended produce (egg/meat) may help clarifying.
Line 266-288: Loss of ability to fly is discussed, however no comparison of the differences between the flying Muscovy duck and the non-flying mallard, in both domesticated Muscovy-mallard and the wild birds, has been presented. Maybe that may help supporting the conclusions.
Line 295: Take care in the use of refereed references. Same issue as in line 71-73 in the introduction. Again taking the length of the domestication age/time span into account may or may not support the conclusions made on the Muscovy ducks ability to fly. Maybe the timeframe for this evolutionary change is too short. A reference to known spatial relation to skeletal changes for comparison will substantiate the conclusions.
Line 310-324: The data handling and analysis of the keel bone observations included in the manuscript is almost absent apart from a “presence-absence” table on the 5 selected criteria, therefore it seems a bit strange that such a large part of the discussion is not based on solid grounds or sufficient data. This is a major weak point. As described above the lack of detailed morphometric measures, which could have been done at the same time as the initial registration, is weakening the results. It should be included, including comparative analysis to the limp data. If not possible, all sections regarding the keel should be taken out of the manuscript. At its present form it does contribute significantly to the rest of the study.

References
Buckner, G.D., Insko, W.M., Henry, A.H., Wachs, E.F., 1949. Rate of Growth and Calcification of the Sternum of Male and Female New Hampshire Chickens Having Crooked Keels. Poult. Sci. 28, 289–292. https://doi.org/10.3382/ps.0280289
Converse, K.A., Kidd, G.A., 2001. Duck plague epizootics in the United States, 1967-1995. J. Wildl. Dis. 37, 347–357. https://doi.org/10.7589/0090-3558-37.2.347
EFSA, 2015. Scientific Opinion on welfare aspects of the use of perches for laying hens. EFSA J. 13, 4131–4171. https://doi.org/doi:10.2903/j.efsa.2015.4131
Eusemann, B.K., Baulain, U., Schrader, L., Thöne-Reineke, C., Patt, A., Petow, S., 2018. Radiographic examination of keel bone damage in living laying hens of different strains kept in two housing systems. PLoS One 13, e0194974. https://doi.org/10.1371/journal.pone.0194974
Fleming, R.H., McCormack, H.A., McTeir, L., Whitehead, C.C., 2004. Incidence, pathology and prevention of keel bone deformities in the laying hen. Br. Poult. Sci. 45, 320–330. https://doi.org/10.1080/00071660410001730815
Letunic, I., Bork, P., 2019. Interactive Tree of Life (iTOL) v4: Recent updates and new developments. Nucleic Acids Res. 47, W256–W259. https://doi.org/10.1093/nar/gkz239
Marchiori, D.M., 2013. Trauma, in: Clinical Imaging: With Skeletal, Chest, & Abdominal Pattern Differentials: Third Edition. Elsevier Inc., pp. 625–765. https://doi.org/10.1016/B978-0-323-08495-6.00010-5
Richards, G.J., Nasr, M.A., Brown, S.N., Szamocki, E.M.G.G., Murrell, J., Barr, F., Wilkins, L.J., 2011. Use of radiography to identify keel bone fractures in laying hens and assess healing in live birds. Vet. Rec. 169, 279-+. https://doi.org/10.1136/vr.d4404
Rufener, C., Baur, S., Stratmann, A., Toscano, M.J., 2018. A Reliable Method to Assess Keel Bone Fractures in Laying Hens From Radiographs Using a Tagged Visual Analogue Scale. Front. Vet. Sci. 5, 124. https://doi.org/10.3389/fvets.2018.00124
Scholz, B., Roenchen, S., Hamann, H., Hewicker-Trautwein, M., Distl, O., Rönchen, S., Hamann, H., Hewicker-Trautwein, M., Distl, O., 2008. Keel bone condition in laying hens: a histological evaluation of macroscopically assessed keel bones. Berl. Munch. Tierarztl. Wochenschr. 121, 89–94. https://doi.org/10.2376/0005-9366-121-89
Swatland, H.J., 1980. Development of Carcass Shape in Pekin and Muscovy Ducks. Poult. Sci. 59, 1773–1776. https://doi.org/10.3382/ps.0591773
Thøfner, I., Hougen, H.P., Villa, C., Lynnerup, N., Christensen, J.P., 2020. Pathological characterization of keel bone fractures in laying hens does not support external trauma as the underlying cause. PLoS One 15, e0229735. https://doi.org/10.1371/journal.pone.0229735
Tracy, L.M., Temple, S.M., Bennett, D.C., Sprayberry, K.A., Makagon, M.M., Blatchford, R.A., 2019. The Reliability and Accuracy of Palpation, Radiography, and Sonography for the Detection of Keel Bone Damage. Animals 9, 894. https://doi.org/10.3390/ani9110894

---

## Round 0.2 · Minor Revisions

Some final helpful presentational and other suggestions have been made by one reviewer. No further review should be required. I will check the final, revised version. Thank you!

Reviewer 1 ·

Basic reporting

the paper is much improved in terms of the writing and sente3nce structure and many of the small errors in logic have been corrected

Experimental design

clarifications to the methods and experimental design have been added

Validity of the findings

sensible conclusions drawn on the main findings

Additional comments

happy with the comments having been addressed appropriately and these have improved the submission

·

Basic reporting

The authors have responded adequately to initial review comments and suggestions and thus really improved the manuscript. There are however still some typos and formatting issues, as well as minor comments and discrepancies in the presentation of the results. Please see specific comments below.

Experimental design

The authors have responded adequately to initial review comments and suggestions

It is acknowledged that the authors are not including broilers, as they are (as mentioned by the authors) juvenile in their intended life span. However their parents, from which they inherit their traits, are fully matures when they reach the end of their normal production cycle (>60 weeks old). Therefore, it would still have been relevant to have had bones from these adult fully matured bird included, as was suggested in the first review of the manuscript. It is however recognised, that those birds, may have been hard to track down in museum collections.

Validity of the findings

The authors have responded adequately to initial review comments and suggestions.

Additional comments

Comments to the author:
Please ensure that you either italicize or not italicize all anatomical names (according to journal policy). At present anatomical names are in italic and not. Both are "legal" ways to write the names according to Smart P., Maisonneuve H. and Polderman A. (eds) Science Editors’ Handbook, European Association of Science Editors. www.ease.org.uk. Section 2.2: Anatomical nomenclature by Jenny Gretton. Here it is also stated that the writing should follow the specific journal policies. Another option could also be to use the common names for the bones/muscles/specific anatomical features.


Specific comments:
The comments below refer to the line numbers in the word version with highlighted changes. Comments to figures and tables refer to the full pdf version of the revised manuscript.

Line 76: comma is missing in reference
Line 79: breed names should be lower cased.
Line 80: semicolon and comma is missing in references
Line 87, 90 & 91: comma is missing in reference
Line 109: Should be Stratmann in reference.
Line 145: anatomical and/or osteological could be added after discrete to emphasize that it is mainly a descriptive analysis of the sternae.
Line 146: This is the first time red junglefowl is mentioned. Shouldn’t it be her the abbreviation is stated for the first time?
Line 163-165 (after …limb.): These two sentences fit better if put after the definition of the total limb length (at the end of the this paragraph).
Line 212-216: It is acknowledged that the authors are not including broilers, as they are (as mentioned by the authors) juvenile in their intended life span. However their parents, from which they inherit their traits, are fully matures when they reach the end of their normal production cycle (>60 weeks old). Therefore, it would still have been relevant to have had bones from these adult fully matured bird included, as was suggested in the first review of the manuscript. It is however recognized, that those birds, may have been hard to track down in museum collections. Please add, information justifying for not using adult fully matured broiler breeder birds (parent stock birds).

Line 219: remove comma after 16
Line 259-262: In the text it is stated that breeds with 4 or more specimen are included to increase rigor. In table 1 and the number of specimens per breed are listed. When looking at this table, there are considerable discrepancies between the in-text listed breeds ≥4 specimens and the listed breeds in the table. Furthermore, kömpfer is not even included in table 1. Additionally, in table 6 and figure 8 it is stated that >5 (which should be replaced with ≥5, to fit data from table 1 accordingly) birds are included in the illustrations. However, neither cochin (n=6) nor seidhenhuhn (n=5) are included, and white leghorn have only 4 specimens.

Line 385: comma is missing in reference
Line 394: remove comma before et al.
Line 397, 398, 400, 402: comma is missing in reference
Line 440: insert space after cases

Figure 5: Is the lines in each box mean or median? What do the error bars present (SD, SEM, CI, interquartiles)? Please update accordingly.
Figure 8: >5 should be replaced with ≥5, to fit data from table 1 accordingly. Is the lines in each box mean or median? What do the error bars present? Please update accordingly. Note the comments for line 259-262 about the missing breeds in the figure.

Table 6: : >5 should be replaced with ≥5, to fit data from table 1 accordingly.

---

## Round 0.3 · accepted · Accept

The reviews were attentively dealt with and so I am happy to recommend that the manuscript is accepted. Congrats!